# Colorectal Cancer Surgery: Influence of Psychosocial Factors

**DOI:** 10.3390/cancers15164140

**Published:** 2023-08-17

**Authors:** Regina Moldes-Moro, María José de Dios-Duarte

**Affiliations:** 1Madrilenian Health Service (SERMAS), 28046 Madrid, Spain; rmoldmor@uax.es; 2Department of Nursing, Faculty of Health Sciences, Alfonso X el Sabio University, 28691 Madrid, Spain; 3Nursing Department, Faculty of Nursing, University of Valladolid, 47005 Valladolid, Spain; 4Nursing Care Research (GICE), University of Valladolid, 47005 Valladolid, Spain

**Keywords:** colorectal cancer surgery, stress, coping styles, illness perception

## Abstract

**Simple Summary:**

Colorectal cancer is one of the leading causes of death in the Western world. The treatment of choice for this disease is surgery. Facing any surgery includes the need for psychological adjustment to the new situation. This study analyses the level of stress that patients experience before potentially curative colorectal cancer surgery, patients’ coping styles and patients’ perception of the disease. This work will help to explore new approaches in the care of these patients and will participate in improving these patients’ states of well-being.

**Abstract:**

(1) Background: In the treatment of colorectal cancer, it is important to consider different psychosocial factors. Our first objective was to measure the levels of perceived stress in subjects diagnosed with colorectal cancer awaiting potentially curative surgery. Also, we aimed to analyse what coping styles these patients used, how they perceived their illness, and the subsequent influence of these factors on their levels of stress. (2) Methods: Stress, coping styles and illness perception were assessed in a sample of 107 patients. The instruments used were the Perceived Stress Scale (PSS-14), the Stress Coping Questionnaire (SCQ) and the Brief Illness Perception Questionnaire (BIPQ-R). (3) Results: Patients using active coping styles have lower levels of perceived stress (*p* = 0.000; *p* = 0.002) than patients making use of passive coping styles (*p* = 0.000; *p* = 0.032; *p* = 0.001). A multi-linear regression model found that the perception of illness and the use of the negative approach coping style (*p* = 0.000; *p* = 0.001) influence an increase in perceived stress, and that a decrease in stress levels was influenced by the problem solving coping style (*p* = 0.001). (4) Conclusions: Based on our results, we recommend preventive interventions in care patients undergoing colorectal cancer surgery.

## 1. Introduction

Cancer is a major public health problem worldwide as it is one of the leading causes of death. In almost all developed countries, cancer is considered the second leading cause of death after cardiovascular disease. In developing countries, the same trend is beginning to appear [1]. 

Colorectal cancer encompasses all tumours located in the large intestine from the ileocecal valve to the rectum, including the appendix. It is one of the leading causes of death in the Western world and in the industrialised nations of Europe, America, Asia, and Australia. Developed countries show a higher incidence than developing countries [2,3].

Colorectal cancer is a treatable and potentially curable disease. Treatment of colon cancer is determined by several factors: tumour stage, anatomical location of the tumour and the patient’s baseline situation. The treatment of choice is surgery, and this manages to cure the disease in approximately 40–45% of those affected. In some cases, five-year survival rates are as high as 65–75%, depending on stage, age or associated complications [1,4].

During the different phases of cancer, patients present a high prevalence of psychopathological disorders and symptoms of anxiety, stress, depression, and emotional distress. These disorders are related to various medical, physical, psychological, and social factors. The prevalence of these disorders among cancer patients is higher than in the general population [5,6,7,8]. 

In men, colorectal cancer is the third leading cancer in terms of incidence and mortality, preceded by prostate and lung cancer [2]. In women, this type of cancer is the second most common, preceded only by breast cancer. The particular characteristics of cancer make it a stressful event that requires a process of adaptation and readjustment [9].

Psychological stress is defined as a particular relationship between the person and the environment, and which the person assesses as threatening or overtaxing his or her resources and endangering his or her well-being [10]. The individual characteristics of the person, the environment, and related factors such as coping or adaptive capacity to precursor stimuli must be taken into account. Thus, for the same type of stimulus, different people will react with different stress responses. Assuming that the situation is not resolved, the organic function may be damaged and some of the organs may be irreparably affected [11,12]. 

Several studies have made the relationship between stress and the development of diseases very clear [13,14], as well as triggering new outbreaks in Crohn’s disease [15]. Stress can therefore be considered a predisposing, triggering or coadjutant factor of illness. Stress is also associated with certain medical treatments and procedures. Thus, surgery is understood by the patient to be a stressor [16,17,18]. 

In addition, stress can alter the function of the immune system in a way that may influence the development or growth of neoplastic diseases [19]. This psychosocial factor is also related to lower survival rates in patients diagnosed with cancer. A review reported that stress generates profound effects on physiological functioning, and this may influence the course of cancer. Stress results in compensatory biological changes to cope with the demands to which the organism is subjected. Thus, the targeted mobilization of resources or their potential depletion incapacitates the organism to some extent to fight effectively against cancer cells. Given the relationship between the neurochemical, hormonal and immune systems, a perturbation in any of these processes could ostensibly increase the proliferation of cancer cells [20]. On the other hand, it should be emphasised that stress favours the development of neoplasia, not only by disrupting immune regulation, but also by damaging DNA and altering repair mechanisms [21]. It should also be noted that psychological stress negatively affects the outcome of cancer treatments [22]. It is therefore important to consider this variable in both the treatment of cancer as well as in the disease itself.

In situations considered dangerous or threatening, people develop coping mechanisms to alleviate unwanted consequences. Thus, people experience the event, analyse the related information, and compare it to the outcome of a previous experience and use this coping mechanism to face new situations. 

Coping is defined as the cognitive and behavioural efforts developed by a person to manage situations they consider to be threatening [10]. Coping includes all consciously generated efforts developed in situations that require mobilization of personal resources to cope with such situations [23]. Thus, on the one hand, there are adaptive behaviours that are triggered when demands exceed the person’s resources, and on the other, coping mechanisms generated by people to face the situation [10]. 

The aim of coping is to control, manage, eliminate, or lessen the adverse impact of stressful situations [24]. This process of dealing with such situations includes different types of behavioural, thought, or emotional responses. This is how coping styles emerge.

Coping styles may be problem-focused and solution-focused, or emotion-focused. The former are also called active coping styles and try to act on the problem if it is perceived as a challenge or threat. The second are also known as passive coping styles, and these make use of emotional management to maintain psychological balance. Thus, within active coping styles we find problem solving, positive reappraisal and seeking social support, whilst within passive coping styles, we find the negative approach, emotional expression, avoidance and religion. Considering the classification of Skinner et al. [25], coping styles are described as follows:

Problem solving involves logical analysis, strategy, planning, and decision making. It turns out to be adaptive because it reduces stress and improves adaptation to the situation.

Positive reappraisal involves the most positive imagination of the problem and is associated with optimism. It is related to better overall health and is considered adaptive.

Seeking social support focuses on seeking contact with other people, asking for advice, and support. It is an adaptive style and reduces the impact of stress on people.

The negative approach implies seeing the situation as uncontrollable. It is related to pessimism and is maladaptive because it leads to lower social adjustment and a higher rate of depression. 

Emotional expression is related to exercising self-control, stoicism, and acceptance of the situation. Generally, it should be considered maladaptive, although sometimes it can be adaptive.

Avoidance involves mental disconnection, denial, and the consumption of food, drugs, or alcohol. It is maladaptive, although it initially dampens the stress response.

Finally, there is religious coping, which involves participation in religious events, parties, and rituals. It seems to provide greater overall well-being, although this is not always the case.

Passive coping styles are associated with poorer adjustment and affective regulation, and higher stress, anxiety, and depression levels, as well as to a loss of control, dependence on others, greater pain, and greater functional impairment [10,26]. Active coping styles are beneficial, reducing stress, anxiety, and depression [10,27]. They are also related to immune response, well-being, and affective states [28,29,30]. 

In relation to coping styles and cancer, active coping styles reduce the impact of cancer and decrease psychological and other psychiatric disorders during the course of the disease. Different studies agree that cancer patients develop different coping strategies based on their emotional state and perception of the illness [31,32,33,34]. 

Greer and Watson [27] showed that active coping styles favour a more successful adaptation process and encourage the person to seek social and informative support, to anticipate potential stressors and to mobilise his or her resources.

In contrast, passive coping styles (maladaptive), increase psychosocial morbidity and increase physical symptoms that may hinder the diagnosis or progression of the disease [24]. 

A coping study carried out among 278 women with cancer showed that the most used strategies were positive and adaptive, particularly in terms of fighting spirit [35]. The outstanding coping styles were those focused on the problem, such as problem solving and seeking social support.

Therefore, coping styles should be considered in cancer, acknowledging that positive coping styles directly affect people with cancer and improve their lives. Being aware that one is vulnerable in life and dealing with this in an active and positive way is very different from using negative coping styles. Coping styles also have an indirect effect on the psychological stress variable, in such that positive coping styles act as stress buffers, facilitating better adaptation and adjustment to the situation. They also reduce stress and thus contribute to slowing down the proliferation of cancer cells. Given the importance of coping styles, it is important to study them in both the treatment of cancer as well as regarding the disease itself.

Illness perception is related to the coping and adaptation mechanisms that a person develops when suffering from an illness, as well as to his or her personal characteristics and sociocultural environment [36,37,38,39]. 

This variable is not influenced by signs and symptoms, but instead by the specific personal significance it has to the person. It should be noted that how an illness is perceived directs the attitude towards it [39]. 

The transactional model of Lazarus and Folkman [10] determined that the perception of illness as a threat will be greater if the person sees the illness as an important or determining factor in his or her life. When a person identifies the pathology as a threat, emotional reactions of fear are triggered by a lack of control over the situation. This circumstance may increase distress levels, anxiety and stress, and the coping styles generated may be less effective.

Some research has established a direct relationship between negative emotions, high stress levels and anxiety, when the illness is perceived as threatening [36,37]. Studies regarding the perception of illness in cancer patients are scarce [40]. It is therefore important to consider this variable both in the treatment of the disease and in the disease itself.

This article helps us to better understand the role of psychological stress, the type of coping style used, and how the illness is perceived in patients with colorectal cancer awaiting treatment with potentially curative surgery. It therefore offers knowledge about how these psychosocial factors and stress can affect patients directly, as well as the stress-generated effects on cancer cells in this disease. In addition, it also provides knowledge regarding how coping styles and disease perception influence stress levels, also contributing to an increase in levels that directly affects cellular changes that harm the DNA and favour metastasis.

Based on the hypotheses that the level of stress in people with colorectal cancer awaiting potentially curative surgery is high, that coping styles and illness perception influence these stress levels, and that these variables are related to each other and influence the presence or absence of post-surgical complications, the aims of this study were as listed below, in order to help improve the treatment and care of these patients:−To measure the level of perceived stress, coping styles and illness perception in these patients.−To analyse and study the relationship between perceived stress and coping styles.−To analyse and study the relationship between perceived stress and illness perception.−To analyse and study the relationship between illness perception and coping styles.−To analyse and study the influence of coping styles and illness perception on the level of perceived stress.−To classify these patients by type, considering the variables studied.−To analyse and study the influence of stress and type of patient, as regards the presence or absence of post-surgical complications.

## 2. Materials and Methods

### 2.1. Participants

A descriptive observational study was carried out which used validated questionnaires to check the influence of stress, coping and the perception of illness in people with colorectal cancer who were to be treated with potentially curative surgery.

The inclusion criterion for this study was being diagnosed with colorectal cancer requiring potentially curative surgery without evidence of metastasis. In terms of tumour staging, all participants were classified as T = 1 N = 0 M = 0 according to the American Joint Committee on Cancer Classification [41]. The exclusion criteria were the presence of psychological afflictions such as obsessive disorders, depression, anxiety, etc. and/or people that required a colostomy during surgery.

The sample was collected between April 2015 and May 2016.

### 2.2. Procedure

Patients were recruited from the Fuenlabrada Hospital’s Surgery Department (Madrid) during the preoperative period for colorectal cancer resection surgery at the time of hospital admission, and after an interview and nursing assessment had been carried out.

In all cases, the research was presented personally, together with an explanation of the criteria necessary to be part of the study. Self-report questionnaires were handed out to those patients who expressed their agreement to participate in the study, once it was clear that they had understood everything correctly.

Participation in this study was disinterested and voluntary and, in all cases, informed consent documents were signed.

The study was approved by the Fuenlabrada Hospital Ethics Committee (Ref. EC828).

### 2.3. Indicators

Three data collection instruments were used to carry out this work along with a sociodemographic questionnaire: the Perceived Stress Scale (PSS-14) by Cohen et al. [42], adapted by Remor and Carrobles [43]; the Ways of Coping Questionnaire (WCQ) by Lazarus and Folkman [10], adapted by Sandín and Chorot [44], which was used as the Stress Coping Questionnaire; and the Brief Illness Perception Questionnaire (BIPQ) by Broadbent et al. [45].

#### 2.3.1. Sociodemographic Questionnaire

This questionnaire explored the age, gender, marital status, and employment situation variables, together with clinical variables such as suture dehiscence, wound infection and paralytic ileus.

#### 2.3.2. Perceived Stress Scale

The PSS-14 was designed to measure the degree to which life situations are evaluated as being stressful. This scale is a self-report instrument that assesses the level of stress perceived during the last month. It consists of 14 items with a five-point scale response format (0 = never, 1 = almost never, 2 = occasionally, 3 = often, 4 = very often). According to the scores obtained, 0 to 14 indicates a low stress level, 15 to 28 indicates a moderate stress level, 29 to 42 indicates an elevated stress level, and 43 to 56 indicates a very high stress level. The Cronbach’s alpha obtained in our study was 827.

#### 2.3.3. Stress Coping Questionnaire

This questionnaire measures 7 basic coping styles: (1) problem solving, (2) the negative approach, (3) positive reappraisal, (4) emotional expression, (5) avoidance, (6) seeking social support and, (7) religion. It is a self-report tool consisting of 42 questions with a graduated Likert-type scale, with 5 response options: never (0), seldom (1), sometimes (2), frequently (3) or, almost always (4). The analysis is based on the sum of the item scores for each coping style. Scores greater than 15 imply that the person possesses that coping style. The Cronbach’s alpha values obtained for the different coping styles were as follows: problem solving = 0.900; positive reappraisal = 0.669; negative approach = 0.868; emotional expression = 0.902; seeking social support = 0.921; avoidance = 0.726; religion = 0.966.

#### 2.3.4. Brief Illness Perception Questionnaire

This questionnaire reveals people’s individual beliefs in five dimensions related to the disease: identity, causes, duration, consequences and controllability. It also allows the global perception of the disease to be known, determining if it is perceived by the person as threatening or not. This is a self-report questionnaire consisting of 8 items with a Likert-type response system ranging from 0 to 10 points. Total values of this scale above 50 indicate perceiving the disease as a threat. The Cronbach’s alpha obtained in our study was 0.799.

All study participants signed informed consent documents and completed the questionnaires in person.

### 2.4. Statistical Analysis

Qualitative variables were analysed as a percentage. An analysis of central tendency measures was carried out for quantitative variables (average) and dispersion (standard deviation).

The reliability of the instruments used was measured using Cronbach’s alpha. This is an index that measures the internal consistency of the scale. In other words, it responds to the average of the correlations between the items it consists of. Cronbach’s alpha = *np*/1 + *p* (*n* − 1), where *n* is the number of items and *p* is the average of all correlations.

To study the relationship between variables, we used Student’s t-test for quantitative variables or one-factor ANOVA, and contingency tables for quantitative and qualitative variables using the chi-square statistic.

A relationship was found between the perceived stress level, coping styles and global illness perception.

Based on the fact that a dependency relationship was established between the perceived stress level, coping styles and global illness perception, a further study of this relationship was carried out. A multi-linear regression analysis was used for this. This statistical technique makes it possible to identify which independent variables can explain a dependent variable, so as to test the causes and to approximately predict the values. The forward stepwise method was used, where the variable that shares the most variance with the criterion variable is introduced first and so on.

Subsequently, a cluster analysis was performed to classify the study participants. This is a multivariate analysis that groups people homogeneously. The aim was to explore how the participants related to each other, taking into account the variables of perceived stress, coping styles and illness perception.

Finally, we wanted to establish the prognosis of postoperative complications, taking into account the level of stress and the classified clusters. For this purpose, a binary logistic regression analysis was used.

The Statistical Package for Social Sciences Version 23 tool (IBM Corp., Armonk, NY, USA) was used to carry out the descriptive statistics and hypothesis contrast of the study, as well as the analyses of the internal consistency, reliability and validity of the instruments used in this work. In all tests, a confidence level of 95% and a *p*-value below 0.05 were considered significant.

## 3. Results

The sample consisted of 107 patients diagnosed with colorectal cancer in preoperative phase, with an average age of 61.94 years, of which 65 were men (60.75%), representing an average age of 61.74 years, and 42 were women (39.25%), representing an average age of 62.26 years.

### 3.1. Stress

The mean of the stress level analysis results for the total sample was 27.19, with a standard deviation of 9.36. The analysis results showed 9 patients with a low stress level (8.41%), 44 patients with a moderate stress level (41.12%), 49 patients with an elevated stress level (45.79%), and 5 patients with a very high stress level (4.67%). Therefore, the predominant stress level in the sample was elevated, followed by moderate. The following table shows the frequency of the perceived stress level and the gender variable (Table 1).

Regarding the relationship between stress and sociodemographic factors, there are statistically significant differences that should be noted between the level of perceived stress (low) in those with an active job (*p* = 0.014). In addition, a statistically significant relationship was found between elevated and very high perceived stress and paralytic ileus (*p* = 0.002).

### 3.2. Coping Styles

With respect to coping styles, we saw that the seeking social support coping style was predominant (46.7%), followed by religion (37.4%), problem solving (37.4%), and positive reappraisal (33.6%) (Table 2).

Statistically significant differences were found for mean age in relation to coping style. The data are shown in Table 3.

As regards problem solving, this coping style is more popularly used by people who are active at work (*p* = 0.007).

With respect to negative approach and marital status (*p* = 0.005), this coping style is less used by couples.

### 3.3. Global Illness Perception

When evaluating the global illness perception, 52 patients (48.60%) did not perceive the illness as a threat, while 55 patients (51.40%) did. Of the patients who did not perceive the disease as a threat, 15.89% were women and 32.71% were men, and in those who did perceive the disease as a threat, 23.36% were women and 28.04% were men.

Regarding illness perception and its relationship with sociodemographic factors, a statistically significant relationship was observed between illness perception and marital status (*p* = 0.011).

Regarding employment and perceiving illness as a threat (*p* = 0.046), the illness perception of housewives was greater.

### 3.4. Relationship between Perceived Stress and Coping Styles

Regarding the results found when relating stress to the different coping styles, the data indicated that there was a statistically significant relationship in the case of some of the coping styles. the results are shown in the following table (Table 4).

Psychological stress was found to be related to the problem solving coping style (*p* = 0.001); the negative approach (*p* = 0.001); positive reappraisal (*p* = 0.002); emotional expression (*p* = 0.032) and religion (*p* = 0.001).

### 3.5. Relationship between Perceived Stress and Global Illness Perception

With regard to the analysis carried out between perceived stress and illness perception, a statistically significant relationship was found between those patients with perceived elevated and very high stress levels and those who also saw their illness as a threat. A statistically significant relationship was also found in patients with perceived moderate stress levels, even though they did not perceive the disease as a threat. The relevant data are shown in the following table (Table 5).

### 3.6. Relationship between Global Illness Perception and Coping Styles

When the analysis between global illness perception and coping styles was carried out, statistically significant differences were found in some coping styles. Table 6 reflects the results found.

### 3.7. Relationship between Coping Styles and Global Illness Perception and Its Influence on the Perceived Stress Level

Following the objectives set and the results obtained, we wanted to assess the influence of coping styles and global illness perception on the perceived stress level in the patients studied. For this purpose, a multi-linear regression analysis was used. The independent variables established were coping styles and global perception of illness, and the dependent variable was perceived stress.

The forward step-wise method was used to perform the multi-linear regression analysis. The variables that demonstrated the strongest relationship with perceived stress were introduced one by one (global illness perception, the problem solving coping style and the negative approach coping style). Table 7 describes the specific statistics of the stepwise model.

The B coefficients that are part of the equation in direct scores are found in the column headed Non-Standardised Coefficients. Thus, the multi-linear regression responds to the following mathematical formula:

Perceived Stress Prognosis = 18.115 + 5.701 × GIP + (−5.354 × CS-PS) + 6.276 × CS-NA).

As for the Beta standardised regression coefficients, they are based on typical scores and are therefore directly comparable to each other. These results indicate the amount of change in typical scores that will occur in the dependent variable for each one-unit change in the corresponding independent variable, when holding all other independent variables constant.

These coefficients provide a very useful clue about the relative importance of each independent variable in the regression equation. In general, a variable has more weight or importance in the regression equation which is higher (in absolute value) than its standardised regression coefficient. When we observe Beta coefficients in Table 8, we realise that the global illness perception variable (0.306) is the most important variable in explaining the perceived stress variable, followed by the negative approach coping style variable (0.281) and lastly the problem solving coping style variable (−0.279).

With regard to statistical significance, we observed that the three independent variables significantly contribute to explaining what is happening with the dependent variable (global illness perception, *p* = 0.000; problem solving coping style, *p* = 0.001; negative approach coping style, *p* = 0.001).

Table 8 summarises the multi-linear regression model for the perceived stress variable, and Table 9 shows the statistical results of the model. Table 8 shows that as the independent variables are introduced, the percentage that explains the variance of the dependent variable increases (R^2^; 229; 0.327; 0.387), so that including the three selected variables explains the variability of perceived stress by 40.4%.

R^2^ expresses the variance proportion of the dependent variable that is explained by the independent variables. At this point, it is necessary to emphasise that this type of relationship does not establish causality; it only establishes the degree of relationship between the dependent variable and the independent variables.

To further refine the calculation of the dependence relationship between the variables introduced in the equation and perceived stress, we use R-squared corrected, which in this case is 0.387. It is interpreted that 38.7% of the variability of the perceived stress is explained by the global illness perception, problem solving coping style, and negative approach coping style variables. R establishes that there is a relationship between the three independent variables and perceived stress.

The multi-linear regression model, which includes these three predictor variables, reduces the prediction errors of the perceived stress score by 38.7%; that is, it explains the percentage of the stress variability. The variables that were finally introduced into the model, and that most directly influence perceived stress, were global illness perception (β = 0.486; *p* = 0.001; R^2^ = 0.229), the problem solving coping style (β = −0.279; *p* = 0.001; ΔR^2^ = 0.098) and the negative approach coping style (β = 0.281; *p* = 0.001; ΔR^2^ = 0.060). The positive or negative value of the Beta coefficient indicates the directionality of the relationship between the variables. Therefore, in this research, it is evident that a higher score in stress was related to a higher score in the global perception of illness as a threat, with less use of the problem solving coping style, and a greater use of the negative approach coping style.

### 3.8. Participant Classification according to Perceived Stress Level, Coping Styles and Global Illness Perception

A cluster analysis was used for this classification. The clusters were calculated on a case-by-case basis. Initially, and to approximate the number of clusters, “none” was selected in the membership cluster. In addition, the dendrogram graph or hierarchical cluster diagram was used. Looking at line 5 of the diagram, three clusters can be seen (Figure 1). The next step was to repeat the analysis by determining the number of clusters we wanted to identify (in our case, three). The results are shown in the table below (Table 10).

The profiles found can be classified as follows:

Profile 1: Study participants with a moderate stress level who use problem solving, positive reappraisal and seeking social support as coping styles, and do not perceive the illness as a threat. These patients use active coping styles.

Profile 2: Study participants with a moderate stress level who use problem solving as a coping style and do not perceive the illness as a threat. These participants primarily use an active coping style.

Profile 3: Study participants with an elevated stress level who use the negative approach, emotional expression and religion as coping styles, and perceive the illness as a threat. These patients use passive coping styles.

### 3.9. Prognosis of Postoperative Complications according to Perceived Stress and Cluster Classification

We wanted to establish the prognosis of post-surgical complications, taking into account the level of stress and the clusters identified. For this purpose, binary logistic regression analysis was used. The binomial dependent variable in both cases was the existence of post-surgical complications with two options: YES and NO. The entry method was used.

In the case of stress, the perceived stress variable was used as the independent variable, and the existence of post-surgical complications was used as the binomial dependent variable.

The main effects model was (χ^2^ = 11.657; df = 1: *p* = 0.001). It was found to be statistically significant.

Therefore, the dependent variable (existence of post-surgical complications) is influenced by the level of stress. 

The variables of the logistic regression model are shown in the following table (Table 11).

For the cluster analysis, this variable was re-coded into dummy variables for logistic regression.

The main effects model for cluster 1 was found to be significant (χ^2^ = 4.781; df. = 1: *p* = 0.029). Therefore, the dependent variable (the presence of postoperative complications) is influenced by belonging to this group.

The variables of the logistic regression model are shown in the following table (Table 12).

The main effects model for cluster 2 (problem solving-moderate stress-no threat) was not significant (χ2 = 0.000; df = 1: *p* = 0.985).

With respect to cluster 3 (passive-high stress-threat), the main effects model was (χ^2^ = 3.876; df = 1: *p* = 0.049). Therefore, the dependent variable (the presence of postoperative complications) is influenced by belonging to this group.

The variables of the logistic regression model are shown in the following Table 13.

## 4. Discussions

This study has helped us to understand the direct impact that perceived stress, coping styles and global illness perception have on colorectal cancer and its surgical treatment, as well as discovering the relationship between these three variables, considering positive coping styles on the one hand, and negative coping styles on the other. It has also shown that both illness perception and the coping styles used have an influence on perceived stress. Positive coping styles contribute to lower levels of perceived stress, and negative coping styles contribute to higher levels of perceived stress.

Regarding perceived stress, the results of our study indicate that a high percentage of the people studied have stress levels ranging from moderate to very high (91%). Taking into account the model of Lazarus and Folkman [10], these stress levels could be related to the patient perceiving the disease and its treatment as a situation that overloads their resources and endangers their well-being. Furthermore, these results are in line with studies showing that surgery itself is a stressor [16,17,18].

Our results show not only that the well-being of these individuals is compromised, but also, given the existing literature, that they are exposed to cellular changes that facilitate the proliferation of cancer cells and thus influence the prevalence of the disease. Furthermore, these stress levels influence the course of the disease, leading to physical deterioration, and are associated with lower survival rates and a poorer response to treatment [19,20,21,22]. It should be noted that in our study, patients who perceived high and very high levels of stress were more likely to develop paralytic ileus (*p* = 0.002).

In general, the predominant coping styles in this research are seeking social support, followed by problem solving, religion and positive reappraisal (Table 3). These are mostly active coping styles. Passive coping styles such as the negative approach and emotional expression are less commonly used, and the least used of all is avoidance. The fact that there are different coping styles in our study is related to the different strategies developed by the patients, influenced in turn by their emotional state and their perception of the disease [31,32,33,34].

Our results show that active coping styles predominate. These contribute to a greater adjustment to the illness and lower stress levels in the patients, due to the beneficial effects of active coping styles on stress [10,46] and hence they improve their psychological situation, the non-proliferation of cancer cells, and enable better management of the disease and its circumstances [19,20,21,22]. 

Regarding illness perception, more than half of the participants considered the disease to be a threatening event. One must remember that this perception is influenced by the personal meaning that the disease has for the person [39]. The fact that these patients perceive the disease as threatening implies a worsening of their emotional and psychological state, which contributes to an increase in stress levels and an increased use of passive coping styles. The latter, in turn, contribute to an increase in the level of perceived stress [36,37]. Our results show that these patients generally perceive the disease as a threat and have higher levels of stress given the direct impact of the disease on stress and its indirect impact through the use of passive coping styles. Passive or maladaptive coping styles increase psychosocial morbidity and increase physical symptoms that may interfere with the diagnosis or the progression of the disease [24]. In this case, illness perception influences the prevalence of colorectal cancer and negatively contributes to the course of the disease, as well as worse treatment outcomes.

With respect to perceived stress and coping styles, there are statistically significant relationships between active coping styles and lower levels of perceived stress, namely problem solving (*p* = 0.000), and positive reappraisal (*p* = 0.002). These results are due to the beneficial effects active coping styles have on decreasing the perceived stress levels of those who use them [10,27]. 

Regarding passive coping styles, statistically significant results were found between the negative approach (*p* = 0.000), emotional expression (*p* = 0.032), and religious coping (*p* = 0.001) and the level of perceived stress. Our results are consistent with those of other research, indicating that negative coping is associated to higher levels of perceived stress [10,46]. 

Regarding the relationship between illness perception and perceived stress (*p* = 0.000), more patients perceived the disease as a threat and presented elevated or very high levels of stress. These results are in line with other published studies [46]. 

In relation to coping and global illness perception, statistically significant relationships were found between the perception of illness as a threat and some coping styles, such as problem solving (*p* = 0.009) and the negative approach (*p* = 0.000).

Our results led us to consider that illness perception together with coping styles have a direct influence upon the perceived stress variable. Thus, it was possible to develop a significant model that establishes the dependence of the perceived stress variable on the independent variables of global illness perception, problem solving and negative approach. It is important to emphasise at this point that a mathematical relationship is established, while in no case is one of causality established.

The development of this model establishes that coping styles are related to the level of perceived stress. Active coping decreases stress and passive coping increases it, as described by Lazarus and Folkman [10]. 

On the other hand, it also establishes that illness perception is related to the level of stress perceived by patients, increasing perceived stress. This has been seen by other researchers in studies carried out regarding other pathologies [30,47].

In addition, the cluster analysis revealed that the people in our sample could be classified into three different types:(1)Those who use active coping styles, have moderate stress levels, and do not perceive the illness as a threat, i.e., active people under moderate stress who are not threatened by the illness.(2)Those who use the problem solving active coping style, have the lowest level of perceived stress, and do not perceive the illness as a threat, i.e., resolute people with moderate stress levels who are not threatened by the illness.(3)Those who use passive coping styles, perceive the illness as a threat, and have the highest levels of perceived stress, i.e., passive people with high stress levels who feel threatened by the illness.

In this classification, active coping styles are related to lower levels of perceived stress, and passive coping styles are related to higher levels of perceived stress.

Taking into account this classification and the multi-linear regression model, it is proven that active coping styles lead to lower levels of stress, with the opposite occurring when passive coping styles are used. Our results are in line with those found in other investigations that clarify the relationship of higher stress levels with passive coping styles [10,26] and that of lower stress levels with active coping styles [10,27]. 

Regarding illness perception, the cluster analysis relates a higher level of perception of the illness as a threat to higher levels of perceived stress. If we take into consideration these results and the multi-linear regression model found, we see that the higher the level of illness perception, the higher the level of perceived stress.

Our findings confirm that stress levels in these patients are influenced by illness perception and coping styles. Passive coping styles are associated with higher stress levels and active coping styles are associated with lower stress levels. At the same time, we can confirm that illness perception also influences very high stress levels. We can also confirm that with regard to the prevalence of colorectal cancer and its surgical treatment, stress worsens the situation of patients. It affects their physical condition, destabilises their emotional state, and negatively influences the course of the disease as well as the patient’s adaptation to it.

Regarding the presence of post-surgical complications, we used binary logistic regression to test whether the presence of complications depended on the level of perceived stress. In addition, cluster 1 (active-moderate stress-no threat) and cluster 2 (passive-high stress-threat) showed statistically significant results regarding the presence of post-surgical complications. Our results are consistent with those found by several authors regarding other types of surgery [47,48]. 

Thus, we can confirm that stress negatively affects the prognosis of surgery and affects the progression of the disease and its treatment.

In view of existing studies and our results, it is very important to point out that, in patients with colorectal cancer awaiting potentially curative surgery, the coping styles they use should be taken into account. Thus, actions aimed at changing coping styles from passive to active should be taken to reduce stress and counteract the perception of the illness as a threat. In turn, the latter would favour a decrease in the level of stress perceived by patients, since both the negative coping style and perception of the disease as a threat lead to elevated and very high levels of stress.

On the other hand, it is important to remember that the latest indications in the management of patients who have undergone colorectal surgery recommend eating and walking sooner to prevent high stress levels occurring [47]. In the experience of doctors, patients who apparently have a higher level of stress are more reluctant to move and eat soon after surgery and tend to stay in bed longer. This may explain the statistically significant relationship we found between paralytic ileus and high stress levels (*p* = 0.002).

In addition, direct actions providing care and assistance based on stress prevention and control would be highly beneficial to patients in the management of their stress.

All this would contribute to improving patient well-being, which in turn would result in shorter stays in hospital and a reduction in hospital admissions.

These types of beneficial actions could be incorporated into pre-surgery care for patients.

## 5. Conclusions

In summary, this study has shown that colorectal cancer patients treated with potentially curative surgery are highly stressed.

Furthermore, it shows that Active coping styles are related to lower levels of perceived stress, while the opposite occurs with passive coping styles. In the case of global illness perception, the same occurs with the type of coping style used. Passive coping styles are related to a higher perception of illness as a threat, while active coping styles are related to a lower perception of illness as a threat.

Moreover, these patients mostly perceive the illness as a threat, and a higher level of illness perception leads to higher levels of stress.

Therefore, we highly recommend that during pre-surgery patient visits, actions are taken to help them manage and reduce their stress levels, as well as direct them towards changing their coping styles from passive to active.

This would help these patients to cope better with the stressful and changing environment of surgical treatment and colorectal cancer itself.

## Figures and Tables

**Figure 1 cancers-15-04140-f001:**
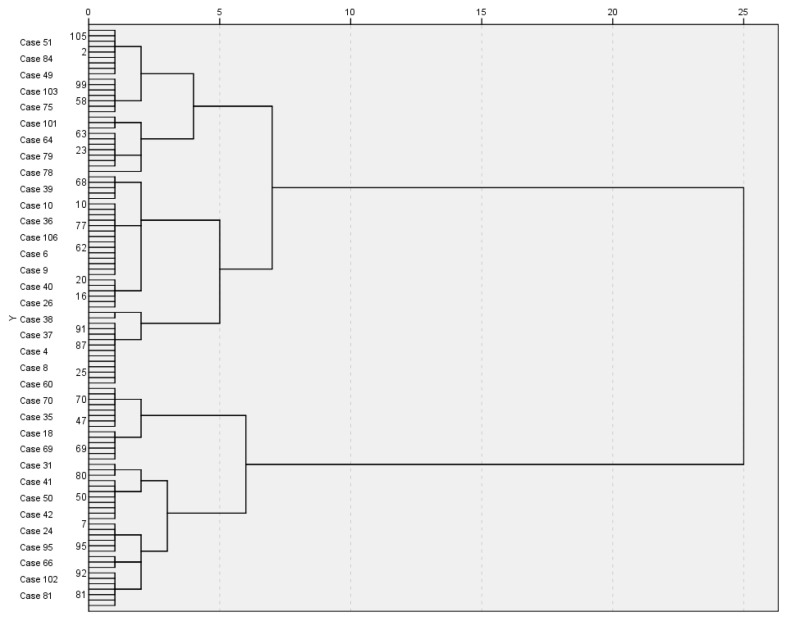
Hierarchical cluster diagram.

**Table 1 cancers-15-04140-t001:** Frequency of each perceived stress level in relation to gender.

Perceived Stress	Gender		
	Male *n* (%)	Female *n* (%)	Total *n* (%)
Low	6 (5.61%)	3 (2.80%)	9 (8.41%)
Moderate	28 (26.17%)	16 (14.95%)	44 (41.12%)
Elevated	27 (25.23%)	22 (20.56%)	49 (45.79%)
Very high	4 (3.74%)	1 (0.93%)	5 (4.67%)

**Table 2 cancers-15-04140-t002:** Frequency of each coping style according to gender.

Coping Style		Gender		
		Male *n* (%)	Female *n* (%)	Total *n* (%)
Problem Solving	Yes	22 (20.56%)	18 (16.82%)	40 (37.4%)
	No	43 (40.19%)	24 (22.43%)	67 (62.26%)
Positive Reappraisal	Yes	21 (19.63%)	15 (14.02%)	36 (33.6%)
	No	44 (41.12%)	27 (22.53%)	71 (66.4%)
Negative Approach	Yes	11 (10.28%)	13 (12.15%)	24 (22.4%)
	No	54 (50.47%)	29 (27.1%)	83 (77.6%)
Emotional Expression	Yes	11 (10.28%)	13 (12.15%)	24 (22.4%)
	No	54 (50.47%)	29 (27.1%)	83 (77.6%)
Seeking Social Support	Yes	37 (34.58%)	22 (20.56%)	50 (46.7%)
	No	28 (26.17%)	20 (18.69%)	57 (53.3%)
Avoidance	Yes	4 (3.74%)	6 (5.61%)	10 (9.3%)
	No	61 (57.01%)	36 (33.64%)	97 (90.7%)
Religion	Yes	22 (20.56%)	18 (16.82%)	40 (37.4%)
	No	43 (40.19%)	24 (22.43%)	67 (62.6%)

**Table 3 cancers-15-04140-t003:** Mean age in relation to coping style.

Coping Style	No		Yes			
	M (DT)	*N*	M (DT)	*n*	*t* (gl)	*p*
Problem Solving	63.6 (9.24)	67	59.18 (12.66)	40	2.08 (105)	0.040
Positive Reappraisal	63.72 (10.30)	71	58.44 (11.06)	36	2.44 (105)	0.016
Negative Approach	61.12 (11.08)	83	64.79 (9.47)	24	−1.47 (105)	0.143
Emotional Expression	61.64 (10.08)	83	63 (13.21)	24	−0.47 (31.15)	0.644
Seeking Social Support	62.14 (9.90)	57	61.72 (11.85)	50	0.20 (105)	0.842
Avoidance	61.87 (10.67)	97	62.7 (12.61)	10	−0.23 (105)	0.817
Religion	61.07 (10.47)	67	63.4 (11.32)	40	−1.08 (105)	0.284

**Table 4 cancers-15-04140-t004:** Relationship between perceived stress and coping styles.

Coping Style	Perceived Stress						
	Low n (%)	Moderate n (%)	Elevated n (%)	Very High n (%)	Total n (%)	X^2^(gl)	*p*
Problem Solving							
No	1 (0.93%)	23 (21.5%)	38 (35.51%)	5 (4.67%)	67 (62.62%)	19.87 (3)	0.000
Yes	8 (7.48%)	21 (19.63%)	11 (10.28%)	0 (0%)	40 (37.38%)		
Positive Reappraisal							
No	5 (4.67%)	21 (19.63%)	40 (37.38%)	5 (4.67%)	71 (66.36%)	14.97 (3)	0.002
Yes	4 (3.74%)	23 (21.5%)	9 (8.41%)	0 (0%)	36 (33.64%)		
Negative Approach							
No	8 (7.48%)	41 (38.32%)	34 (31.78%)	0 (0%)	83 (77.57%)	26.00 (3)	0.000
Yes	1 (0.93%)	3 (2.8%)	15 (14.02%)	5 (4.67%)	24 (22.43%)		
Emotional Expression							
No	7 (6.54%)	39 (36.45%)	32 (29.91%)	5 (4.67%)	83 (77.57%)	8.78 (3)	0.032
Yes	2 (1.87%)	5 (4.67%)	17 (15.89%)	0 (0%)	24 (22.43%)		
Seeking Social Support							
No	5 (4.67%)	30 (28.04%)	20 (18.69%)	2 (1.87%)	57 (53.27%)	7.36 (3)	0.061
Yes	4 (3.74%)	14 (13.08%)	29 (27.1%)	3 (2.8%)	50 (46.73%)		
Avoidance							
No	9 (8.41%)	42 (39.25%)	41 (38.32%)	5 (4.67%)	97 (90.65%)	5.46 (3)	0.141
Yes	0 (0%)	2 (1.87%)	8 (7.48%)	0 (0%)	10 (9.35%)		
Religion							
No	6 (5.61%)	35 (32.71%)	26 (24.3%)	0 (0%)	67(62.62%)	15.74 (3)	0.001
Yes	3 (2.8%)	9 (8.41%)	23 (21.5%)	5 (4.67%)	40(37.38%)		

**Table 5 cancers-15-04140-t005:** Relationship between perceived stress and global illness perception.

Global Illness Perception		Perceived Stress					
	Low n (%)	Moderate n (%)	Elevated n (%)	Very High n (%)	Total n (%)	X^2^(gl)	*p*
No Threat	6 (5.61%)	34 (31.78%)	12 (11.21%)	0 (0%)	52 (48.6%)	31.79 (3)	0.000
Threat	3 (2.8%)	10 (9.35%)	37 (34.58%)	5 (4.67%)	55 (51.4%)		

**Table 6 cancers-15-04140-t006:** Relationship between global illness perception and coping styles.

Coping Style		Global Illness Perception			
	No n (%)	Yes n (%)	Total n (%)	X^2^(gl)	*p*
Problem Solving					
No	26 (24.3%)	41 (38.32%)	67 (62.62%)	6.88 (1)	0.009
Yes	26 (24.3%)	14 (13.08%)	40 (37.38%)		
Positive Reappraisal					
No	30 (28.04%)	41 (38.32%)	71 (66.36%)	3.40 (1)	0.065
Yes	22 (20.56%)	14 (13.08%)	36 (33.64%)		
Negative Approach					
No	49 (45.79%)	34 (31.78%)	83 (77.57%)	16.14 (1)	0.000
Yes	3 (2.8%)	21 (19.63%)	24 (22.43%)		
Emotional Expression					
No	48 (44.86%)	35 (32.71%)	83 (77.57%)	12.63 (1)	0.000
Yes	4 (3.74%)	20 (18.69%)	24 (22.43%)		
Seeking Social Support					
No	34 (31.78%)	23 (21.5%)	57 (53.27%)	5.96 (1)	0.015
Yes	18 (16.82%)	32 (29.91%)	50 (46.73%)		
Avoidance					
No	50 (46.73%)	47 (43.93%)	97 (90.65%)	3.61 (1)	0.057
Yes	2 (1.87%)	8 (7.48%)	10 (9.35%)		
Religion					
No	38 (35.51%)	29 (27.1%)	67 (62.62%)	4.73 (1)	0.030
Yes	14 (13.08%)	26 (24.3%)	40 (37.38%)		

**Table 7 cancers-15-04140-t007:** Statistics and coefficients of the multi-linear regression model for the perceived stress variable.

		Non-Standardised Coefficients		Typified Coefficients		
		B	Standard Error	Beta	T	Sig.
1	(Constant)	13.373	2.530		5.287	0.000
	Global Illness Perception	9.050	1.587	0.486	5.704	0.000
2	(Constant)	24.516	3.636		6.742	0.000
	Global Illness Perception	7.483	1.533	0.402	4.882	0.000
	Problem Solving	−6.384	1.583	−0.332	−4.032	0.000
3	(Constant)	18.115	3.964		4.570	0.000
	Global Illness Perception (GIP)	5.701	1.557	0.306	3.662	0.000
	Problem Solving (CS-PS)	−5.354	1.542	−0.279	−3.472	0.001
	Negative Approach (CS-NA)	6.276	1.878	0.281	3.342	0.001

Note: A. Predictor variables (Constant): global illness perception. B. Predictor variables (constant): global illness perception, problem solving coping style. C. Predictor variables (constant): global illness perception, problem solving coping style and negative approach coping style.

**Table 8 cancers-15-04140-t008:** Summary of the regression model for the perceived stress variable.

Model	R	R-Squared	R-Squared Corrected	Standard Estimation Error
1	0.486	0.237	0.229	8.2024
2	0.583	0.340	0.327	7.6644
3	0.636	0.404	0.387	7.3150

**Table 9 cancers-15-04140-t009:** Regression model statistics for the perceived stress variable.

	β	*F*	R*^2^/*ΔR^2^	*p*
Negative Approach	0.281	23.311	0.060	0.001
Problem Solving	−0.279	26.763	0.098	0.001
Global Illness Perception	0.486	32.536	0.229	0.001

Note: R^2^ is used in global illness perception as a constant variable in the model, and increased R squared (ΔR^2^) for the rest of the model’s variables, in order to not lose the power of the first variable.

**Table 10 cancers-15-04140-t010:** Profile means in the perceived stress, coping styles and global illness perception variables.

Variables		Participants	
	Profile 1	Profile 2	Profile 3
	M	M	M
Perceived Stress	24.2	21.5	34.3
Problem Solving	15.7	15.5	7.3
Positive Reappraisal	16.7	13.2	8.5
Negative Approach	8.2	5.7	15
Emotional Expression	5.6	5	15.02
Seeking Social Support	16.2	8.8	13.7
Avoidance	10.7	4.8	10
Religion	12	4.5	15
Global Illness Perception	45.3	45.4	59.2

**Table 11 cancers-15-04140-t011:** Binary logistic regression model for the prediction of post-surgical complications considering perceived stress.

	B	*E.T.*	*Wald*	*df*	Sig.
Perceived Stress	0.076	0.024	10.332	1	0.001
Constant	−1.987	0.668	8.848	1	0.003

**Table 12 cancers-15-04140-t012:** Binary logistic regression model for the prediction of post-surgical complications considering cluster 1 (active-moderate stress-no threat).

	B	*E.T.*	*Wald*	*df*	Sig.
Active-Moderate Stress-No Threat	−0.995	0.467	4.549	1	0.033
Constant	−0.302	0.226	1.786	1	0.181

**Table 13 cancers-15-04140-t013:** Binary logistic regression model for the prediction of post-surgical complications considering cluster 3 (passive-high stress-threat).

	B	*E.T.*	*Wald*	*df*	Sig.
Passive-High Stress-Threat	−0.794	0.408	3.780	1	0.050
Constant	0.550	0.324	2.878	1	0.090

## Data Availability

Not applicable.

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
