# Peer review of "Colorectal Cancer Surgery: Influence of Psychosocial Factors"

_cancers, 2023, doi:10.3390/cancers15164140_

Round 1

Reviewer 1 Report

Dear all,

I suggest  to reconsider the manuscript. Specifically : 

Introduction 

Authors descripte findings of previous studies but it isn't cleary What the current study contribuites to this literature. 

I would suggest eleborating on the main purpose of this study, mnntioned only in the abstract but not in the manuscript .

It's not cleary how this study will help to explore new approach in the care of these patients. 

Discussion

The authors should include a discussion ofhow the findings from the current study fit into the literature on the prevalence 

minor editing of English language required 

Author Response

We thank the reviewer for taking the time to review our article.

Your sincerely,

The Authors

Reviewer 2 Report

In the manuscript, the author analyzes the level of stress that patients present before potentially curative colorectal cancer surgery, patients' coping styles and patients' perception of the disease. By questionnaire survey, the authors evaluated the stress, coping style and illness perception of 107 patients. The research has the potential to explore new approaches in the care of these patients and improve their health. But they still need to be improved as mentioned below.

1.      The formula and calculation process of Cronbach's alpha should be listed in the manuscript.

2.      Does the preoperative stress level affect the prognosis of patients?

3.      The prognosis level of patients grouped by cluster analysis should be compared.

4.      Will the results of the questionnaire be affected by the degree of the patient's illness?

5.      What is the influence of different Coping styles on Perceived Stress?

6.      The number of samples included in the cluster analysis should be stated.

Author Response

(The authors gave the same response as above.)
